# Cytoprotective Effects of Taurine on Heat-Induced Bovine Mammary Epithelial Cells In Vitro

**DOI:** 10.3390/cells10020258

**Published:** 2021-01-28

**Authors:** Hui Bai, Tingting Li, Yan Yu, Ningcong Zhou, Huijuan Kou, Yingying Guo, Liang Yang, Peishi Yan

**Affiliations:** 1College of Animal Science and Technology, Nanjing Agricultural University, Nanjing 210095, China; bbbbbhhhww@163.com (H.B.); litingtingnjau@foxmail.com (T.L.); 13382049293@126.com (Y.Y.); 13436110762@163.com (Y.G.); yangliangagu@sina.com (L.Y.); 2Ulanqab Animal Husbandry Workstation, Ulanqab 012000, China; znc888@126.com (N.Z.); kouhuijuan0113@126.com (H.K.)

**Keywords:** MAC-T cells, taurine, heat stress, mitochondrial damage, oxidative stress, cell apoptosis

## Abstract

It is a widely known that heat stress induces a reduction in milk production in cows and impairs their overall health. Studies have shown that taurine protects tissues and organs under heat stress. However, there have yet to be studies showing the functions of taurine in mammary alveolar cells-large T antigen (MAC-T) (a bovine mammary epithelial cell line) cells under heat shock. Therefore, different concentrations of taurine (10 mM, 50 mM, and 100 mM) were tested to determine the effects on heat-induced MAC-T cells. The results showed that taurine protected the cells against heat-induced damage as shown by morphological observations in conjunction with suppressed the translocation and expression of heat shock factor 1 (HSF1). Moreover, taurine not only reversed the decline in antioxidase (superoxide dismutase (SOD) and glutathione peroxidase (GSH-PX)) activities but also attenuated the accumulation of malondialdehyde (MDA). Meanwhile, mitochondrial damage (morphology and complex I activity) resulting from heat exposure was mitigated. Taurine also alleviated the rates of cell apoptosis and markedly depressed the mRNA expressions of BCL2 associated X, apoptosis regulator (BAX) and caspase3. Furthermore, compared with the heat stress (HS) group, the protein levels of caspase3 and cleaved caspase3 were decreased in all taurine groups. In summary, taurine improves the antioxidant and anti-apoptosis ability of MAC-T cells thereby alleviates damage of cells due to heat insults.

## 1. Introduction

With the global temperature continuously rising, the negative impact of the greenhouse gas emissions on animal husbandry is increasingly intensified [1]. Heat stress, as a major environmental stressor, brings major challenges to the dairy industry [2], with previous studies reporting that heat stress not only decreases the milk production but also impairs the health of cows, including by increasing somatic cell counts and metabolic disorders occurrence [3,4]. Therefore, heat stress causes great economic loss to the dairy industry [5,6]. Meanwhile, the metabolism of mammary epithelial cells is also changing dramatically [7]. In one study, it was found that, compared with a thermoneutral group, there were 2716 markedly differently expressed genes in the bovine mammary epithelial cells of the heat-treated group [8]. Accumulating evidence indicates that heat stress is associated with a change in cell junction [9], decreasing cell viability, increasing oxidative stress, and increasing cell apoptosis [10]. Therefore, it is of vital significance to identify effective means to mitigate the damage caused by heat stress.

A variety of research has been directed toward enhancing the functions of the mammary gland under high ambient temperatures to steadily maintain the lactation of the mammary epithelial cells [11,12]. Taurine is recognized to be a beta-amino ethanesulfonic acid, which can be found to exist as a free state in all animal tissues. Moreover, taurine has many biological and pharmacological functions [13] and is considered to be a strong antioxidant. Extensive research has shown that taurine improves antioxidant levels, such as superoxide dismutase (SOD), glutathione peroxidase (GSH-PX), as well as catalase (CAT) [14,15]. Previous studies have also focused on the importance of taurine in terms of its anti-apoptosis effect, with one study indicating taurine markedly hampered increases in the expression of Bax, Caspase9, Caspase3, Cytochrome c (Cyt-c) and P53 in broiler aortic endothelial cells under heat shock [16]. Furthermore, the taurine additive has been reported to have anti-inflammatory effects [17,18], as well as treating intractable diseases, such as Alzheimer’s disease [19]. Thus, taurine appears to be an attractive candidate for use as a cytoprotector and heat-induced mitigators; therefore, it seems to be a reasonable hypothesis that taurine may attenuate heat-induced dysfunction in cow mammary epithelial cells.

However, to the best of our knowledge, no report has yet documented any effects of taurine in mammary alveolar cells-large T antigen (MAC-T) cells under heat exposure. Therefore, the aim of this study was to investigate the effects of taurine supplementation on the expressions of genes and proteins related to oxidative stress and apoptosis under control and heat stress conditions in order to investigate the roles of taurine and its mechanism of action in mitigating heat-induced damage caused to MAC-T cells.

## 2. Materials and Methods

### 2.1. Chemicals

Taurine (suitable for cell culture, meets USP testing specifications) was purchased from Sigma Chemical Co. (St. Louis, MO, USA).

### 2.2. Cell Culture and Treatment

MAC-T (a bovine mammary epithelial cell line) cells (a gift from associate researcher G. Xing, Jiangsu Academy of Agricultural Sciences) were incubated in Dulbecco’s modified Eagle’s medium (DMEM)/Ham’s F-12 medium (Hyclone, Logan, UT, USA) with 10% fetal bovine serum (FBS) (ScienCell, Carlsbad, CA, USA) and 200 U/mL of penicillin and streptomycin (Hyclone, Logan, UT, USA). The cells were cultured at 37 °C in a humidified atmosphere containing 5% CO_2_.

The cells were divided into five groups through randomization: A control (C) group, heat stress (HS) group, low taurine (HS + LTau) group, moderate taurine (HS + MTau) group, and high taurine (HS + HTau) group. After the cells reached 70% confluence, prior to heat treatment, they were cultured for 24 h under DMEM/F12 with serum-free conditions. Then, the HS group was cultured at 42.5 °C for one hour and recovered at 37 °C for another 12 h; the taurine groups were pretreated with 10 mM, 50 mM, 100 mM of taurine for two hours followed by culturing at 42.5 °C for one hour and recovery at 37 °C for 12 h. The C group was continuously cultured in 37 °C and received no taurine treatment.

### 2.3. Cell Viability

Cell viability was quantified by 3-(4,5-dimethylthiazol-2-yl)-2,5-diphenyltetrazolium bromide (MTT) medium (KeyGEN BioTECH, Nanjing, China). The MAC-T cells were seeded in 96-well plates prior to incubating for one hour or not at 42.5 °C; then, the cells were recovered at 37 °C for different amounts of time. Additionally, the cells were treated with various concentrations of taurine (0, 10, 50, 100, and 200 mM) for two hours and then cultured for another 12 h. According to the instruction, 50 μL of the 1 × MTT solution (1:4 diluted with dilution buffer) was added to each well, and the cells were maintained for four hours at 37 °C containing 5% CO_2_. Then, 150 μL of dimethyl sulfoxide (DMSO) (Sigma, St. Louis, MO, USA) was added, and the plates were oscillated before measuring the OD at 550 nm using a microplate reader.

### 2.4. Ultrastructural Observation

The MAC-T cells were seeded into six-well plates and treated as described above. Then, 2 mL of 2.5% glutaraldehyde fixative was added into each well. The cells were scraped gently and collected into the centrifuge tubes. Then, the samples were removed from the fixing solution and rinsed three times with phosphate buffer (PBS) before administering 1% osmium tetroxide for postfixing. After dehydration, infiltration, and embedding, the samples were cut into ultrathin (60–80 nm) sections and stained with uranyl acetate and lead citrate. Then, the samples were observed with a transmission electron microscope (HT7700, Hitachi, Tokyo, Japan).

### 2.5. Detection of Mitochondrial Complex I Activity

Mitochondrial complex I activity was estimated via the colorimetric method using commercial kits (Nanjing Jiancheng Bioengineering Institute, Nanjing, China). The determination consisted of three steps: Background control determination, total activity determination, and non-specific activity determination. Mitochondrial proteins were extracted with a mitochondrial protein extraction kit (Nanjing Jiancheng Bioengineering Institute, Nanjing, China), and complex I activity was quantified in the mitochondria.

### 2.6. Antioxidant Capacity Measurement

The cells were cultured in six-well plates for 48 h prior to heat-induced treatment; then, they were treated and harvested. The oxidative parameters (SOD, GSH-PX, and MDA) were detected using the corresponding commercial kits (Nanjing Jiancheng Bioengineering Institute, Nanjing, China). Furthermore, the protein concentration was detected using a protein assay kit (Nanjing Jiancheng Bioengineering Institute, Nanjing, China) in accordance with the manufacturer’s instructions. Then, we used a microplate reader to measure the absorbance of each well.

### 2.7. Detection of Apoptosis

Cell apoptosis was assessed using an Annexin V-fluorescein isothiocyante (FITC) /propidium iodide (PI) apoptosis detection kit (Absin Biochemical Company, Shanghai, China). According to the manufacturer’s recommendation, the cells were treated as indicated before being harvested and then washed twice with cold PBS. Following this, 300 μL of 1 × binding buffer (1:9 diluted with binding buffer) was added to suspend the cells. The cells were stained with 5 μL of Annexin V-FITC and 5 μL of PI in each tube. Before the analysis, we added 200 μL of the 1 × binding buffer to each well. Finally, cell apoptosis was analyzed with a FACS Calibur (BD Biosciences, Bedford, MA, USA) flow cytometer (FCM), and 1 × 10^4^ cells were detected for each tube at minimum. The results were analyzed using FlowJo software.

### 2.8. RNA Isolation and Quantitative Real-Time PCR (qPCR)

The total RNA was extracted with a Trizol reagent (Invitrogen, Carlsbad, CA, USA) and reversed transcribed with a PrimeScript RT reagent (Takara, Tokyo, Japan). The purity of the total RNA and cDNA was measured by a nanodrop spectrophotometer. Primers were designed by the National Center for Biotechnology Information (NCBI) Primer-BLAST and synthesized by a commercial corporation (Sangon Biotech, Shanghai, China), as shown in Table 1. Ribosomal protein S15 (RPS15), ribosomal protein S9 (RPS9), and ubiquitously expressed prefoldin like chaperone (UXT) were used as housekeeping genes. The mixtures (cDNA products, premix EX Taq, primers, ROX, and water) were performed in a quantstudio 5 real-time PCR instrument (Applied Biosystems, Carlsbad, CA, USA) using a SYBR premix EX Taq kit (Takara, Tokyo, Japan). The 2^−ΔΔCt^ method was applied to calculate the relative mRNA abundance.

### 2.9. Immunofluorescent Staining

The cells were cultivated in six-well plates 48 h before treatment. After heat exposure, they were washed two times with PBS and fixed in 1 mL 4% paraformaldehyde for 15 min. After permeabilizing and blocking, primary antibody: Heat shock factor 1 (HSF1) (dilution 1:200, Proteintech, Wuhan, China) was added, and the samples were incubated for 12 h at 4 °C. After three washes with PBS, the cells were incubated with a secondary antibody for 50 min at room temperature. Then, after washing again, the nucleus were stained with 4′,6-diamidino-2-phenylindole dihydrochloride (DAPI) for 10 min. Subsequent imaging and analysis of the samples were performed using a fluorescence microscope (Zeiss, Oberkochen, Germany).

### 2.10. Western Blotting Assay

The protein expressions of HSF1, heat-shock protein 90 (HSP90), caspase3, and cleaved caspase3 were examined via western blot assay. The cells were lysed with RIPA buffer for 30 min on the ice and centrifugated at 12,000 rpm for 10 min at 4 °C. Then, the supernatants were collected as the total protein solution. Proteins were determined using a BCA protein quantification kit (Nanjing Jiancheng Bioengineering Institute, Nanjing, China) according to the manufacturer’s instructions. Proteins were denatured by heating. Then, the lysates were separated, transferred, and blocked by 5% milk. In accordance with the antibody instructions, membranes were incubated overnight at 4 °C with the following primary antibodies: HSF1 (1:1000, Proteintech, Wuhan, China), HSP90 (1:2000, Proteintech, Wuhan, China), caspase3 (1:400, Proteintech, Wuhan, China). The membranes were washed three times, and then the secondary antibodies were added for 30 min and diluted at a ratio of 1:3000. The protein bands were detected with a luminescent image analyzer (Fujifilm, Tokyo, Japan) using an enhanced chemiluminescence (ECL) substrate (Thermo Scientific, Waltham, MA, USA). Finally, western blotting results were quantified using ImageJ software (National Institutes of Health, Bethesda, MD, USA).

### 2.11. Statistical Analysis

All measurements were performed in triplicate, and all data was expressed as the mean ± standard error of the mean (SEM). Statistical analysis was performed with one-way ANOVA followed by the least significant difference (LSD) method for multiple comparisons. All statistical tests were carried out by SPSS 21.0 (IBM, New York, NY, USA), with *p* < 0.05 considered as statistically significant. * *p* < 0.05, ** *p* < 0.01, and *** *p* < 0.001.

## 3. Results

### 3.1. Cell Viability and Change in Cell Morphology of Mammary Epithelial Cells after Heat-Induced Treatment

In order to explore the most appropriate time point for treating the MAC-T cells, the cells were exposed to heat shock for one hour, and recovered at 37 °C for different durations. As shown in Figure 1A, the cell viability gradually decreased within 12 h recovery time, causing about a 40% decrease in cell viability (*p* < 0.001). However, the cell viability almost fully recovered at 24 h. In addition, when the cells were treated with 0–200 mM taurine (Figure 1B) for two hours followed by cultivation for another 12 h, no significant impact of the cell viability was observed up to 200 mM taurine (Figure 1C). Hence, we treated the cells at 12 h recovery time with three different doses of taurine (10 mM, 50 mM, and 100 mM) to investigate the effects of taurine on heat-induced MAC-T cells.

Cell morphology was also observed with an inverted light phase-contrast microscope. Cells in the C group had a normal shape. In contrast, some cells exhibited abnormal cell junction and shape in the HS group, which coalesced into a mass. Interestingly, 50 mM and 100 mM taurine protected the morphology of the MAC-T cells against heat-induced alteration (Figure 1D).

### 3.2. Taurine Alleviates Heat Shock Response in MAC-T Cells

Moreover, we researched whether taurine played a role in the heat-induced treatment that triggered the heat shock response of MAC-T cells. As shown in Figure 2A, the heat shock response of the cells was activated for essential survival under heat exposure. The transcriptions of HSF1 and HSP90 were significantly enhanced. Then, we studied the translocation of HSF1 from the cytoplasm to the nucleus, and Figure 2B indicated that cells in the C group exhibited low levels of HSF1, while the cells in the HS group showed higher levels of HSF1 in both the cytoplasm and the nucleus, and taurine suppressed the translocation of HSF1. In addition, the mRNA expression levels and protein levels of HSF1 were increased in the HS group compared with the C group, while they were markedly decreased in all taurine groups. The change in the mRNA expression abundance of HSP90 was consistent with that of HSF1; however, the protein expression of HSF90 was not significant between groups (Figure 2E).

### 3.3. Taurine Relieves Structural Damage and the Decline in the Complex I Activity of Mitochondria under Heat Shock

In addition, to further study the function of taurine on the intracellular structure of heat-induced MAC-T cells, ultrastructural alterations were observed using a transmission electron microscope. As shown in Figure 3A, the C group displayed a complete structure and rich cristae in the mitochondria. The mitochondrial morphology was markedly changed under heat stress, with observations such as the swollen structure of the mitochondria with the loss of the cristae in the HS group. Furthermore, 10 mM and 50 mM of taurine effectively attenuated the structural damage done to the mitochondria. Moreover, mitochondrial complex I activity was suppressed (*p* < 0.05) by heat exposure, and was markedly elevated in the HS + LTau group (*p* < 0.05) (Figure 3B).

### 3.4. Taurine Enhances Antioxidant Capacity of Heat-Induced MAC-T Cells

To investigate whether taurine can affect the antioxidant capacity of MAC-T cells induced by high-temperature, some antioxidant parameters were detected in this study. Our results showed that the MDA level was significantly higher in the HS group than in the C group (*p* < 0.01), and the activities of SOD and GSH-PX were reduced in the HS group compared to in the C group (*p* < 0.01). The MAC-T cells in the taurine groups exhibited lower levels of MDA than in the HS group (*p* < 0.01) (Figure 4B). In addition, there was no marked difference in the SOD activity between the HS and HS + LTau groups; however, the activities of SOD in the HS + MTau group and HS + HTau group were significantly higher than in the HS group (Figure 4A). Similarly, pretreatment with 10mM taurine reversed the activities of GSH-PX in the HS-induced MAC-T cells, while the other two groups showed no effect (Figure 4C). In summary, heat stress associated with oxidative damage was effectively declined by taurine.

### 3.5. Taurine Reduces Cell Apoptosis Induced by Heat Shock

We next investigated whether taurine could also relieve the level of cell apoptosis in heat-induced MAC-T cells. The rates of cell apoptosis were detected with an Annexin V-FITC/PI detection kit, and the results were shown in Figure 5A. The ratio of cell apoptosis increased markedly when the MAC-T cells were treated with heat shock (*p* < 0.001). The 10 mM, 50 mM, and 100 mM of taurine treatments all significantly alleviated cell apoptosis in the heat shock-induced MAC-T cells. In addition, compared to the HS group, the mRNA abundance of BCL2 associated X, apoptosis regulator (BAX) (a pro-apoptosis marker), caspase3 (a pro-apoptosis marker), and the ratio of BAX/BCL2 apoptosis regulator (Bcl-2) were markedly decreased in all taurine groups (Figure 5C,E,F), while there was no significant difference with the C group. Furthermore, compared with the HS group, the protein expressions of caspase3 and cleaved caspase3 declined with all taurine treatments—but at varying degrees (Figure 5G). Therefore, taurine may play an anti-apoptotic role by rescuing cells under heat exposure.

## 4. Discussion

Earth’s increasing temperature has had a major impact on the dairy industry, including reducing productivity [20,21], increasing metabolic disorders prevalence, and even enhancing mortality [22]. Furthermore, heat stress induces oxidative stress, apoptosis, and other series of events in cow mammary epithelial cells in vitro [23,24]. Many studies have focused on identifying additives that can relieve the damage caused by heat shock. Taurine was first isolated from ox bile, which has a wide variety of biological and pharmacological activities [13]. Many previous studies have shown taurine as a kind of antioxidant [25], a nutritional supplementation [26], that is strong when focused on minimizing the severity of various diseases caused by oxidative damage and many kinds of stress [19,27,28]. Although there are many studies on taurine, there have been none—until now—investigating its molecular mechanism in bovine mammary epithelial cells under heat stress. This study was thus the first to clarify the effects of taurine pretreatments in heat-induced MAC-T cells, which were cultivated at 42.5 °C for one hour and recovered at 37 °C for 12 h.

Previous cell culture-based studies have investigated that the cell viability of heat-induced bovine mammary epithelial cells (BMECs) is the lowest when cells are recovered for 12 h at 37 °C [24], which is consistent with the results of our present study. Furthermore, it has been indicated that a considerable additional dosage of taurine has been used from 1 to 100 mM concentrations [29]. In this study, we used 0–200 mM of taurine to study taurine’s effects on cell viability and found that pretreatment with taurine became toxic when the concentration reached 200 mM. Therefore, this study was designed to study the dosages of 10 mM, 50 mM, and 100 mM.

When cells survive under heat exposure, their heat shock response is activated. The overexpression of heat shock protein (HSP) is triggered by HSF1. HSF1, which separates with HSPs (HSP90 for the most part), enters the nucleus and induces the expression of downstream heat shock element regulatory genes [30]. Our data indicated that the heat stress upregulated the mRNA expression abundance of HSF1 and HSP90. With the different recovery times, the expression level fluctuated [31]. Furthermore, the translocation of HSF1 from the cytoplasm to the nucleus also increased in the MAC-T cells under heat shock. In contrast, all the different taurine supplementation levels reduced the translocation of HSF1 and suppressed the mRNA abundance and protein levels of HSF1. However, taurine had no effect on the protein expression of HSP90. These findings were consistent with the study of Belal in terms of the role of taurine in the mRNA and protein expression levels of HSP90 in liver tissue [32]. Future studies are needed to identify the levels of other HSPs under heat stress and further determine the effects of taurine. The present study indicated that taurine improved the thermotolerance of the MAC-T cells by reducing the translocation and the protein level of HSF1 in order to prevent the expression of downstream heat shock element regulatory genes in heat-induced MAC-T cells.

It is widely accepted that mitochondrial functions depend on the integrity of mitochondria. Mitochondrial calcium influx is triggered when it suffers an attack. Then, the mitochondrial nucleases are activated to indiscriminately degrade all mitochondrial polynucleotides [33]. In this study, ultrastructural observation showed that heat exposure could damage the mitochondrial structure, but, when the cells were pretreated with different concentrations of taurine, this damage was relieved. Moreover, the mitochondrial complex I activity in the HS + LTau group was elevated to resist mitochondrial dysfunction. These results were supported by those of Zhuang, studying in chronic heat-stressed broilers [34], and Jong, studying cardiomyocytes [35].

Meanwhile, the present study provided insight into the role of taurine as an antioxidant. The steady-state concentrations of pro-oxidants and antioxidants are disturbed, which induces oxidative stress in cells [36]. Traditionally, SOD and GSH-PX are the main antioxidant defense systems in cells and are associated with the antioxidant capacity of cells [37]. Heat stress has been considered to decrease the activities of SOD [23,24] and GSH-PX [34], while MDA (a last product of lipid peroxidation) levels have been increased under heat shock in many previous studies [23,24,38]. Consistently, taurine can rescue oxidative stress by restoring the activity of SOD and decreasing the level of MDA in the present study. Interestingly, compared with the HS group, taurine significantly decreased the levels of GSH-PX only in the HS + LTau group [39], which suggested that it perhaps depended on the taurine dosage. Several studies have proposed that taurine decreases oxidative stress by forming a conjugate (5-taurinomethyluridine) with a key uridine moiety in the wobble position of mitochondrial tRNALeu (UUR), the deficiency of which leads to the defective assembly of respiratory chain complex I, resulting in the cell undergoing oxidative stress; this then indicates that taurine played a vital role in attenuating the oxidative stress [35,40].

It is demonstrated that heat stress can trigger oxidative stress [23,24], and exposure to oxidative stress induces a wide series of responses, such as apoptosis and, finally, necrosis [41,42]. In the present study, the taurine pretreatment groups exhibited significantly declined the rates of apoptosis when compared with the HS group. The mRNA expression levels of BAX and caspase3 were markedly lower than those of the HS group. Moreover, the ratio of BAX/Bal-2 was decreased significantly. Furthermore, the protein levels of caspase3 and cleaved caspase3 in the HS + LTau, H S+ MTau, and HS + HTau groups were decreased when compared with those of the HS group. These results were line with the studies of Kai Liu in porcine kidney-15 cells [43] and Yan Li in PC12 cells [29]. Based on the above investigations, it could be concluded that taurine directly rescued mammary epithelial cells by decreasing the rates of apoptosis in order to maintain the capacity of the mammary glands to synthesize and store milk.

## 5. Conclusions

In general, heat stress significantly enhanced the translocation and expressions of HSF1 and the mRNA abundance of HSF90, to the point that a heat stress response was triggered. Furthermore, heat exposure–induced damage done to the mitochondria as well as oxidative stress were activated in the MAC-T cells. Then, exposure to oxidative stress induced the cell apoptosis. On the one hand, taurine effectively inhibited the translocation of HSF1 and the mRNA and the protein levels of HSF1 in order to improve the thermotolerance of the MAC-T cells. On the other hand, taurine relieved damage done to the mitochondrial structure and the decline in complex I activity, thereby rescuing the antioxidants activities. Finally, taurine reversed the rates of apoptosis in the HS-induced MAC-T cells (Figure 6). Therefore, this study provided useful evidence that taurine, as a novel natural additive, could possibly be used in cow feed under heat exposure.

## Figures and Tables

**Figure 1 cells-10-00258-f001:**
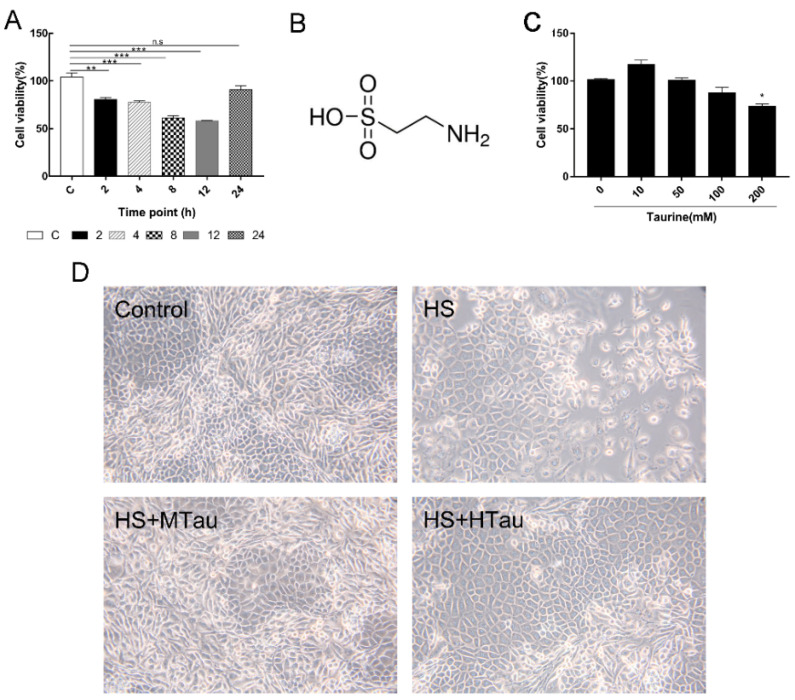
Effects of taurine supplementation on the cell viability and cell morphology of mammary alveolar cells-large T antigen (MAC-T) cells. (**A**) cell viability induced by heat shock on MAC-T cells, (**B**) chemical structure of taurine, (**C**) cell viability when treated with different concentrations of taurine for two hour and then recovered for another 12 h, and (**D**) phase-contrast micrographs of MAC-T cells were exposed to heat stress with and without taurine. Magnification 200×. * *p* < 0.05, ** *p* < 0.01, and *** *p* < 0.001.

**Figure 2 cells-10-00258-f002:**
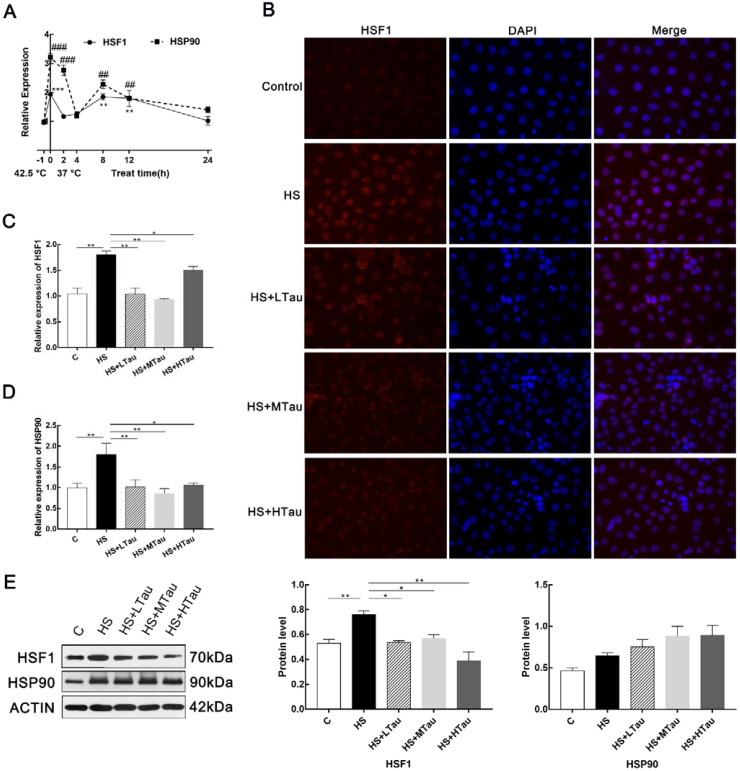
The role of taurine in MAC-T cells in response to heat shock. (**A**) the transcriptions of heat shock factor 1 (HSF1) and heat-shock protein 90 (HSP90) changes with the recovery time, (**B**) the translocation of HSF1. Magnification 400× (**C**) HSF1 mRNA levels, (**D**) HSP90 mRNA levels, and (**E**) HSF1 and HSP90 protein levels. All data are presented as the mean ± SEM. * *p* < 0.05, ** *p* < 0.01,*** *p* < 0.001, ## *p* < 0.01, ### *p* < 0.001.

**Figure 3 cells-10-00258-f003:**
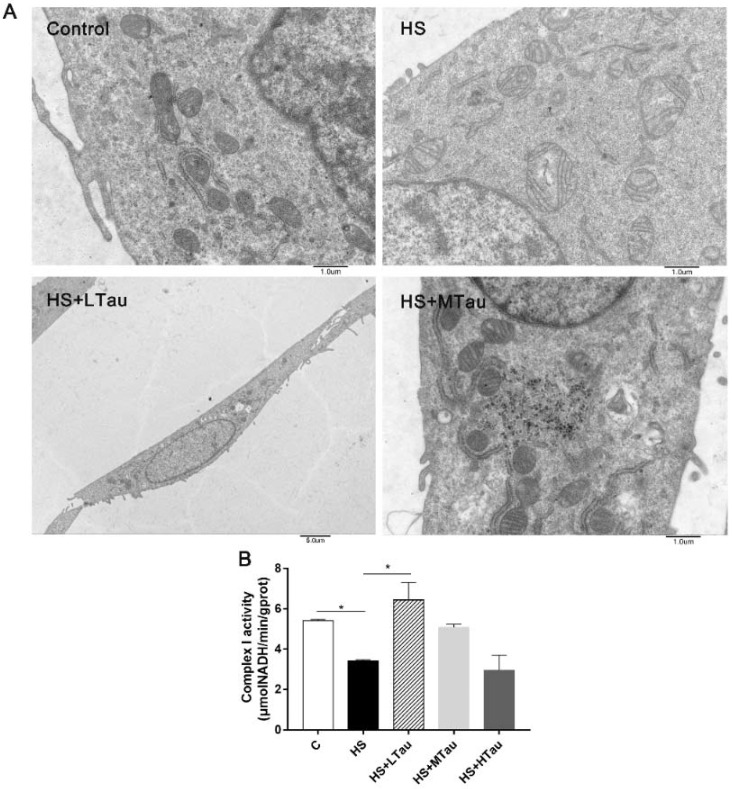
Taurine supplementations relieve the damage done to the mitochondrial structure and the decline in complex I activity in MAC-T cells under heat exposure. (**A**) the mitochondrial structures and (**B**) mitochondrial complex I activity. Transmission electron microscopy images at original magnifications of ×8000 Control, heat stress (HS), and moderate taurine (HS + MTau), and ×2000 low taurine (HS + LTau). * *p* < 0.05.

**Figure 4 cells-10-00258-f004:**
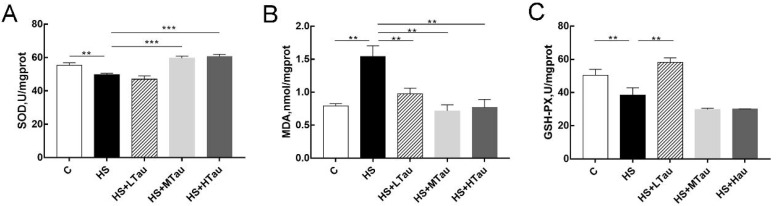
Taurine improved the activities of antioxidant enzymes and the antioxidant capacity of heat-induced MAC-T cells. (**A**) the activity of superoxide dismutase (SOD), (**B**) malondialdehyde (MDA) production, and (**C**) glutathione peroxidase (GSH-PX) activity. ** *p* < 0.01, and *** *p* < 0.001.

**Figure 5 cells-10-00258-f005:**
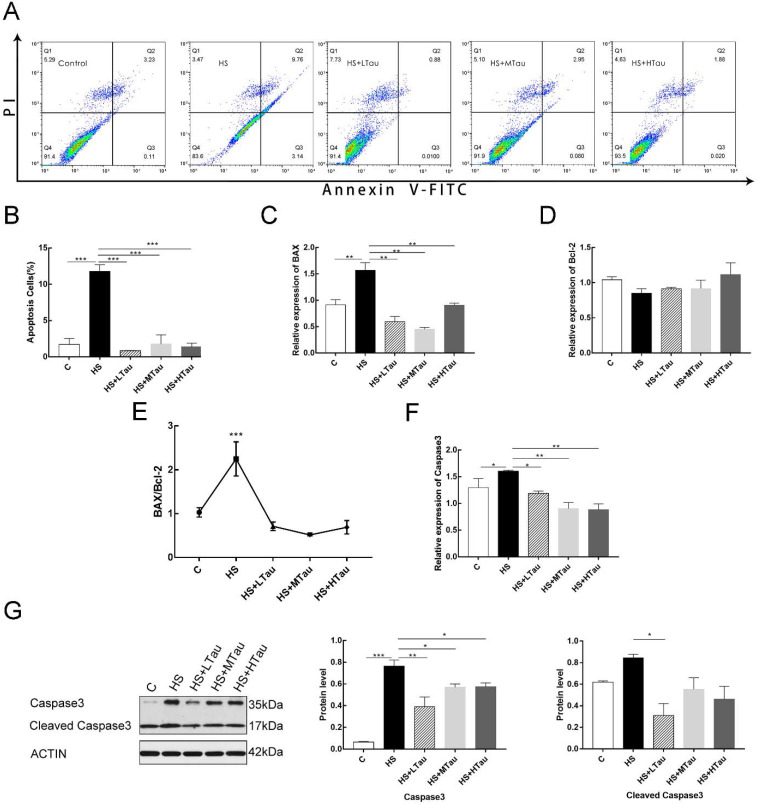
Taurine pretreatments attenuated cell apoptosis in MAC-T cells under heat exposure. (**A**) effects of taurine on apoptosis in MAC-T cells induced by heat shock analyzed via flow cytometry. (**B**) cell apoptosis rates. (**C**) mRNA abundance of BCL2 associated X, apoptosis regulator (BAX). (**D**) mRNA abundance of BCL2 apoptosis regulator (Bcl-2). (**E**) the ratio of BAX/Bcl-2. (**F**) mRNA abundance of caspase3. (**G**) the protein expression of caspase3 and cleaved caspase3 were determined by Western blotting. All data are presented as the mean ± SEM. * *p* < 0.05; ** *p* < 0.01; *** *p* < 0.001.

**Figure 6 cells-10-00258-f006:**
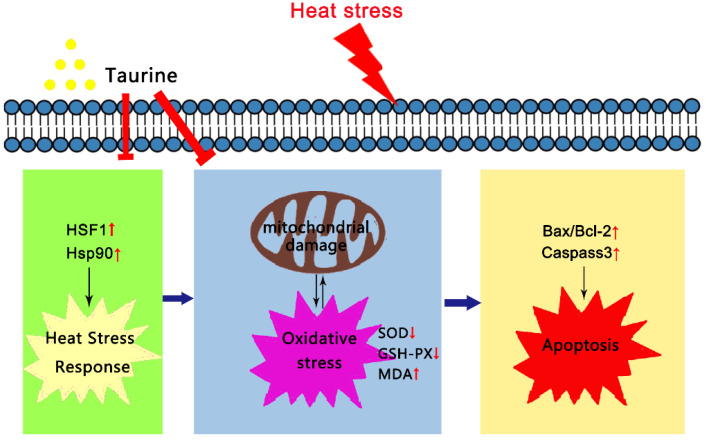
Taurine protected heat-induced MAC-T cells effectively by improving the thermotolerance of cells and reducing the levels of oxidative stress and cell apoptosis.

**Table 1 cells-10-00258-t001:** Sequences of primers for the real-time PCR assay.

Genes	GenBank Number	Primer Sequence (5′-3′)
RPS9	NM_001101152.2	Forward: TCTTGGTTTCCAGAGCGTTG
		Reverse: ATACTCGCCGATCAGCTTCA
UXT	NM_001037471.2	Forward: CGCTACGAGGCTTTCATCTCT
		Reverse: CGAGTGGTTAGCTTCCTGGAG
RPS15	NM_001024541.2	Forward: CAAGATGGCGGAAGTGGAAC
		Reverse: GTAGCTGGTCGAGGTCTACG
BAX	NM_173894.1	Forward: CTGAGCGAGTGTCTGAAGCG
		Reverse: ACAGCTGCGATCATCCTCTG
BCL2	NM_001166486.1	Forward: AGGCTGGGACGCCTTTG
		Reverse: GGGCTTCACTTATGGCCCAG
Caspase3	XM_015473877.2	Forward: TGGCGAAATGCAAAGAACGG
		Reverse: TGTGAGCGTGCTTTTTCAGC
HSF1	NM_001076809.1	Forward: AGCACGCCCAGCAACAGAAAG
		Reverse: CCGCCGTCGTTCAGCATCAG
HSP90	NM_008302.3	Forward: GATGGAAGAGGAGGAGGTGGAGAC
		Reverse: AGGGCGTCAGACGAGTTTGAAATC

## Data Availability

Restrictions apply to the availability of these data.

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
