# Peer review of "Cytoprotective Effects of Taurine on Heat-Induced Bovine Mammary Epithelial Cells In Vitro"

_cells, 2021, doi:10.3390/cells10020258_

Round 1

Reviewer 1 Report

This is an interesting study on the effects of taurine on heat-induced bovine mammary epithelial cells in vitro.

The methodology seems to be appropriate including study design, experimentation, and result interpretation. Graphical presentations are also of very good quality and very easy to understand. Moreover, statistics are very well performed. The statistical software cited is unclear. Is it SPSS or StatSoft? The main problem of this manuscript is poor English language. Too many grammar and syntax errors make the manuscript difficult to follow. The authors are strongly advised to seek professional language editing. Some of these errors are presented below:

Title: in vitro as well as all Latin words and expressions should be written in italics.

Line 9: induces instead of induce

Line 11: a study which/that shows or showing instead of shows

Line 12: suffering from

Line 13: the effects

Line 18: The comma after apoptosis is not needed

Line 21: antioxidant

Line 32: with the thermoneutral group

Line 37: towards enhancing functions of the mammary..

Line 42: the colon after as should be omitted

Line 44: a taurine additive was found or was reported

Line 47: dysfunction

Line 49: has documented any effects

Line 50: determine/investigate instead of research the

Line 65: a space is missing after the colon

Line 69: recovered at instead of in

Line 78: maintained

Line 85: and rinsed three times with buffer…

Line 101: of PI in each tube

Line 127: prior to adding the secondary antibodies

Line 129: FUJIFILM

Line 148: the effects

Line 150: a normal shape

Line 159: play a role in …. that triggers

Line 160: The heat shock

Fig. 2 increasement does not exist

Line 171: by a … microscope

Line 180: of the

Line 181: the antioxidant

Line 182: the same

Line 185: groups and group

Line 186: marked difference

Line 189: remove comma after while

Fig 4: improved the activities of heat-induced

Line 217: that can

Line 222: there are no

Line 226: heat-induced

Line 228: Furthermore, …. a considerable

Line 229: the effects

Line 233: the heat

Line 248: omit of after from

Line 249: insights into

Line 258: in the present study

Line 261: function

Line 264: can trigger

Line 265: stress induces… finally necrosis

Line 268: the ratio

Line 269: Furthermore

Reviewer 2 Report

Major concerns

  1. Grammars should be corrected thoroughly and writings should be improved. Here are examples in Abstract

Ex. line 9, widely accepted that heat stress induces a reduction in milk production and impairs the health in cows.

Ex. line 11-12, However, there is no study showing the functions of taurine in MAC-T (a bovine mammary epithelial cell line) cells under heat shock.

Ex. line 12, Therefore, different treatment concentrations of taurine (10

13 mM, 50 mM, 100 mM) were tested to determine its effects on heat-induced MAC-T cells. 

Ex. line 14, The results showed that, taurine protected the cell against heat-induced damages as shown by morphological observations in conjunction with significantly suppressed mRNA levels of HSF1 and HSP90 expressions.

Ex. line 20, as compared with the HS group.

Ex. line 20-22, In summary, taurine improves antioxidat and anti-apoptosis ability of MAC-T cells and thereby alleviates damages of cells against heat insults.

  1. The authors need to justify why BMECs were used a model in the study. Any in vivo studies or with primary cells?
  2. How the authors concluded that only HSF1 and HSP90 mediate the cytoprotective effects of taurine? Any genetic intervention by siRNA or pharmacological inhibitor treatments ? Also, most of the results are real-time PCR analyses, at least HSF1 and HSP90 results should be presented with Western blots results? A functional assay such as HSF translocation in gel shift analysis may be required.
  3. The time course of HSF1 and HSP90 expression should be examined again, since cell functions such as death comes out after a variety of gene expression networks. Usually, most cells respond to treatment with mRNA expression after 3 hrs and with protein changes after 5-6 hrs.
  4. Figure 3, the authors need to show at least a function assay with mitochondria such as complex activity in electron transport chain, not merely in morphology.
  5. The authors need to explain the mechanisms of taurine to promote anti-oxidative activities, by the antioxidant activity of taurine itself per se, or through other pathway to influence gene expression in anti-oxidative defense?      

Round 2

Reviewer 2 Report

  1. Grammars should be corrected thoroughly and writings should be checked and improved again. Here are examples in Abstract and Introduction

Line 12, Studies have been yet conducted to shown

Line 18-20,  Meanwhile, mitochondrial damages (morphology and complex â…  activity) resulting from heat exposure were mitigated.

Line 22, and cleaved

Line 23, and thereby

Line 33-34, including increased somatic cell counts and metabolic disorders occurrence.

Line 36, is also changed.

Line 38, Accumulating evidences indicate…

Line 39, decrease of cell viability, increase of oxidative stress and cell apoptosis.

Line 46, Extensive researches have shown

Line 48 in respect to its..

Line 49, as one study showed markedly hampered increases by taurine treatment in the expression of Bax,…..

Line 51, , taurine as a dietary supplement has been reported to have anti-inflammatory effects [17, 18] and as a treatment in intractable diseases, such as Alzheimer’s disease [19] .

Line 53, mitigators. Therefore, it is very plausible to hypothesize that taurine….

Line 56, documented regarding to any effect of taurine on…

Line 60, damage in MAC-T cells.

  1. In Figure 1 and 2, what is control (C), before heat stress or a parallel without heat stress to 24 hr?
  2. Figure 4 title, “Taurine improves the….” and Figure 5 title “Taurine pretreatments attenuate…”